# Impact of dietary patterns, individual and workplace characteristics on blood pressure status among civil servants in Bida and Wushishi communities of Niger State, Nigeria

**Phillips Edomwonyi Obasohan** [1]*, **Josephine N. Okorie**[2]ᵒ, **Adamu Lapai Sule**[3]ᵒ, **Khadijat Jumai Ndako**[2]ᵒ

1 Department of Liberal Studies, College of Administrative and Business Studies, Niger State Polytechnics, Bida Campus, Bida, Niger State, Nigeria, 2 Department of Hospitality Management, College of Administrative and Business Studies, Niger State Polytechnics, Bida Campus, Bida, Niger State, Nigeria, 3 Department of Library Information Science, College of Administrative and Business Studies, Niger State Polytechnics, Bida Campus, Bida, Niger State, Nigeria

ᵒ These authors contributed equally to this work.
* philiobas@yahoo.com

**Data Availability Statement:** All relevant data are within the manuscript and its Supporting Information files.

## Abstract

The global burden estimate of hypertension is alarming and results in several million deaths annually. A high incidence of sudden deaths from cardiovascular diseases in the civil workforce in Nigeria is often reported. However, the associations between Dietary Patterns (DPs), individual, and workplace characteristics of hypertension among this workforce have not been fully explored. This study aimed to identify DP in the Bida and Wushishi Communities of Niger State and establish its relationship with hypertension along with other individual and workplace characteristics. Factor analysis was used to establish DP, Chi-square test to identify their relationships with hypertension, and logistic regression to determine the predictor risk factors. The prevalence of hypertension was 43.7%; mean weight, height, and body fat were: 72.8±15 kg, 166±8.9 mm and 30.4%, respectively. Three DPs: "Efficient Diet," "Local diet," and "Energy Boost Diet" were identified. The factor loading scores for these factors were divided into quintiles Q1–Q5; none of them had a significant effect on hypertension status. Conversely, increase in age, the Ministry, Department, and Agency (MDA) of employment, frequency of eating in restaurants, and obesity were identified as significant risk factors. After adjusting for confounders (age, body mass index, MDA, and eating habits), a high score (Q5) in "efficient diet pattern" was significantly related to a lower likelihood of hypertension than a low score (Q1). The prevalence of hypertension among the participants was relatively very high. An increase in age and working in educational sector were risk factors associated with hypertension. Therefore, it is recommended that civil servants engage in frequent exercise and undergo regular medical checkups, especially as they get older. These findings highlight the need for large-scale assessment of the impact of variables considered in this study on hypertension, among the civil workforce across Niger state and Nigeria.

**Funding:** This research was funded by TetFund Nigeria via Institution-Based Research (IBR) to Niger State Polytechnic with grant number: TETFUND/DESS/POLY/ZUNGERU/2016/VOL.1. The grant was jointly given to the authors. The funding agency had no role in the design of the study; in the collection, analyses, or interpretation of data; in the writing of the manuscript, or in the decision to publish the results.

## 1. Introduction

Non-communicable Diseases (NCDs), especially hypertension (HBP), have become major globalized health risks[1] in recent times [2] with the burden becoming higher in developing countries [3]. The global burden estimate of hypertension is well over 1 billion people, resulting in about 7.1 million deaths per year[4,5]. The danger is that it is increasingly underdiagnosed, inadequately treated, and inadequately controlled, especially in most urban cities of Sub-Saharan Africa (SSA) [6]. The high prevalence of HBP in Nigeria, of more than 48% among the adult population[7], is an alarming burden to public health in Africa as it is the most populous country in the region with over 180 million people[8]. Now with rising epidemiological, demographic and lifestyle changes, the expectation is in continuous increase[7,9].

Many factors are considered to be associated with hypertension. The Framingham study [10], as many others [11,12], found that age is directly correlated with both systolic blood pressure (SBP) and Diastolic Blood Pressure (DBP) in men and in women. Studies, however, have reported inconsistencies pertaining to gender disparity in HBP. For instance, a recent study showed that men have a significantly lower probability of hypertension than women at stage 1, but not at stage 2[13]. Adeoye et al [14] observed that some reports have shown that women have poorer blood pressure control than do men, while others claimed they have equal or better control than do men. Furthermore, gender disparities in HBP have been shown to be age-dependent[15]. High prevalence HBP rates of 49% and 47% have been observed in Nigerian adult males and females, respectively[7]. These gender disparities in HBP patterns are also observed for many other countries[15,16]. Significant associations between blood pressure variables, weight and body mass index (BMI) have also been reported in several studies [16,17]. BMI is regarded in other studies in Nigeria as the gold standard for defining overweight and obesity[18,19], and an indicator for overall adiposity [19,20]. Other studies have also shown that obesity increases the likelihood of having hypertension by 3.5-fold[21,22].

Studies on diet and nutrition have spanned over and above the prevention of deficiency-related diseases to prevent chronic non-communicable diseases including hypertension [23,24]. Earlier studies have identified several dietary patterns and their association with colorectal cancer, type 2 diabetes, anemia, and breast cancer, among several regionally representative samples, in America and China [25–29]. Some healthful dietary patterns were discovered to be effective in lowering blood pressure and preventing hypertension[30]. For instance, the "fruit and milk" pattern was found to be associated with a lower prevalence of hypertension among adult Chinese men [31]. However, many studies investigating how dietary patterns relate to disease burden have used single foods and food groups resulting in inconsistent results because of a high degree of correlation among dietary constituents[32]. One major resultant problem here is that single foods or nutrients do not act in isolation but rather combine their effects. A more robust approach is suggested, one that would capture the total dietary experience, including all the nutrient interactions. Factor analysis can help to achieve this.

In Nigeria, a limited number of studies have used factor analysis to investigate diet pattern as a risk factor for non-communicable diseases. However, the patterns of these associations with hypertension among the workforce in Nigeria have not received attention. In this study, therefore, we have used principal component factor analysis to identify dietary patterns and subsequently categorize the factor scores to determine their association with lifestyle; food habits; workplace status; and other socioeconomic and demographic variables on how they impact on blood pressure of the civil workforce in the Bida and Wushishi communities in Niger State, Nigeria.

## 2 Materials and methods

### 2.1 Study population and location

The study population comprised civil servants working in government ministries, departments, and agencies (MDA) in the Bida and Wushishi communities of Niger state. The two communities in focus were purposely selected such that Bida, a Local Government Area in Niger State was taken as an urban area, while Wushishi, another Local Government Area was considered a rural area ensuring urban-rural mix in the sample. The synergy between these two communities' workforce is that some of the government tier workforces in the state have representation, and the workforce members in these communities interact frequently with each other. In addition, no significant difference in hypertension status was observed among the civil workforce members by place of residence in these communities.

### 2.2 Sampling procedure

The survey was conducted as a community-based cross-sectional descriptive survey. Eligible civil servants were interviewed using an interview-administered Food Frequency Questionnaire (FFQ) which was designed following guidelines provided by the Food and Agricultural Organization (FAO) and used elsewhere [33]. The questionnaire had 5 sections: (i) socioeconomic, workplace, and demographic data (SWD); (ii) lifestyle and physical activities pattern (LPA); (iii) food habits (FH); (iv) dietary diversity and frequency (DDF); and (v) health condition/anthropometric data (HCA. The reliability of this instrument was confirmed using the test re-test method, which resulted in correlation coefficient of 0.92.

A starting MDA was determined at random. All eligible participants who were active, serving, civil servants in any of the 3-tier places of work (Federal, State, and Local Government), between the ages of 15 and 65 years, residing in any of the study communities and who consented to the survey were interviewed and had their data collected. Pregnant and lactating women and physically challenged civil servants were excluded from participating in the study [20]. In situations where we were unable to find the total number of participants needed for the survey in any MDA, the nearest MDA was visited until the required respondents were found. The survey was conducted over a period of 6 weeks (from July 3rd to August 14th, 2018) by the 4 co-authors and assisted by 3 trained undergraduates. English language was the medium of communication during the survey. However, where it became necessary to explain to the participants in local dialects, this was also done.

### 2.3 Sample size determination

Studies have reported that to be able to estimate a prevalence between 10–50% with a 5% desired precision, design effect of 2, and 5% type 1 error, a minimum sample size of 400 will be needed [5,34]. Adjusting for a non-response rate of 10% gave a minimum sample size of 440. In this study, we administered 530 questionnaires, 497 (93.8%) were returned and captured. A total of 21 (3.96%) were rejected for improper filling, bringing about a sample size of 476 participants (i.e., 89.8% response rate) which was used for the study.

### 2.4 Ethical approval

The project was approved by the Research and Development Committee of Niger State Polytechnic, Zungeru. Permission to collect data from subjects was obtained from the Administrative Heads of the MDAs and oral informed consent was obtained from all respondents.

## 2.5 Variable measurements and operational definitions

The dependent (outcome) variable was the Resting Pressure Status of participants measured on a continuous scale (both Systolic and Diastolic readings were recorded) using standard procedures and an OMRON apparatus (Model M3, HEM-7131-E). The subject was made to sit down for at least five minutes prior to testing. Their left arm was bare and resting on a table at an angle of 45 degrees with the palm facing upwards. A cuff of appropriate size was wrapped firmly around the wrist, the start button was pressed and the cuff inflated. Once maximum inflation was reached the cuff deflated automatically and both the resting blood pressure and the resting pulse rate were recorded. The hypertension status was assessed following the updated WHO/ISH classification of Hypertension guidelines [35] when Systolic Blood Pressure is 140 mm Hg and above or/and Diastolic Blood Pressure is 90 mm Hg and above as cut-offs.

The predictor (independent) variables considered were: self-reported age of respondents at the time of survey, classified as: "1" for 15–24, "2" for 25–34, "3" for 35–44, "4" for 45–54, and "5" for ≥55 years of age; gender of respondents: classified as "1" for male and "2" for female; government tier place of work, categorized as "1" for federal employees and "2" for state, and "3" for local government employees and MDA (Ministries, Department and Agencies) place of work to include: education classified as "1," health as "2," finance as "3," agriculture "4", rural development as "5" and others (these include all other MDAs that could not be classified among the ones identified) as "6." Other variables investigated included: place of residence (rural "1" or urban "2"); self-reported education status as "1" for "not educated" (having primary education and below) and "2" educated (having secondary education and above); marital status; religion status and income status. Data on lifestyle and physical activities pattern (LPA) were collected around 3 major activities: smoking, alcohol and physical exercises, with each respondent stating the status (for instance, self-reported smoking status was categorized as "1" for "current smoker", "2" for "ex-smoker", and "3" for "never smoked"; if participant currently smoked, the frequency is categorized as "1" for "1 stick/day," "2" for "2 sticks/day" and "3" for "3 sticks/day", and "4" for "more than 3 sticks/day"; and duration of smoking was categorized as "1" for "once or twice a week", "2" for 3 or 4 times a week", and "3" for more than "4 times a week"); food habits (FH) and dietary diversity and frequency (DDF) which had 44 food items. For each of the food items, participants were asked how often they consumed them over the past 12 months (daily, once a week, 2–3 times a week, 4–6 times a week, occasionally or never). We further grouped the food items into 7 predefined food groups according to their similarities[30]. The body mass index, which was measured on a continuous scale, was obtained using a full-body sensor body composition monitor and scale (Omron model HBF-516, Omron Healthcare, USA) [36] after the height (measured with tape-rule on the wall following the standard procedure) and self-reported age were inputted into the scale. The WHO cut-offs for BMI were used to categorize BMI (under-weight ($<18.5$ kg/m$^2$) classified as "1," normal weight (18.5–24.9 kg/m$^2$) classified as "2," over-weight (25–29.9 kg/m$^2$) classified as "3" and obese ($>29.9$ kg/m$^2$) classified as "4"). The scale also estimated the body fat and skeletal muscle percentages, and visceral fat by the bioelectrical impedance method [26]. The procedures were carried out by trained personnel.

## 2.6 Statistical analysis

The following data analysis methods were adopted:

1. From the FFQ, items that had theoretical negative and positive effects on outcome variables were assigned scores ranging from "0" to "5" and "5" to "0," respectively [37]. To determine

the dietary pattern, the composite score method[38] was used to compute scores for each food group. This resulted in a global score ranging from 0 minima, for vegetable and fruit food groups, to 39 maxima recorded in cereal products. Principal Component Factor (PCA) of Factor Analysis (FA) with varimax orthogonal rotation was used for greater interpretation and for a more simplistic structure[30]. What the factor analysis (or principal component analysis) does is to examine the correlation matrix of food variables and search for underlying traits (or factors) that explain most of the variation in the data. Thus, a large number of food variables are reduced to a smaller set of variables that capture the major dietary traits in the population[25]. The sample adequacy to justify the use of FA was tested with the Kaiser–Meyer–Olkin (KMO) and Bartlett test of Sphericity to establish the presence of correlations between variables[23,39]. Factor loading was determined for the extracted factors using the Kaiser-criterion of cut off ± 0.3 [23,39].

2. To examine the relationship between selected principal independent variables with hypertension status, we used the chi-square test with 95% confidence interval. Factors that were found significant at 5% level from the bi-variable analysis were further analyzed using a multiple logistic regression model for categorical outcomes to determine the likely effects of the selected factors on hypertension status of civil servants in Bida and Wushishi communities using STATA version 14 for academic users [40] for the computations.

## 3 Results

### 3.1 Anthropometric variables and dietary pattern

Table 1 shows that the prevalence rate of Hypertension (HBP) among civil servants in Bida and Wushishi communities was 43.7%. Also, 24.3% of the respondents were obese and another 32.7% were over-weight. The mean weight, height and body fat for all respondents were: 72.8 kg (±15 kg), 166 mm (±8.9 mm) and 30.4% respectively.

Principal Component Analysis (PCA) with varimax orthogonal rotation method was used to extract three factors of dietary patterns. Prior to rotation, these factors accounted for 66.7% of the total variation. The Bartlett test of sphericity was significant rejecting the hypothesis that the variables were not intercorrelated. A KMO of 0.695 signified that the sample was adequate for factor analysis [39]. Factor 1 was characterized by foods such as nuts and legumes, animal products, fats and oil, vegetables and fruits. This was labeled "Efficient Diet" and accounted for 36.3% of the total variance. Factor 2 was characterized by, cereal products, nuts, and legumes. These are foods commonly grown in these communities, were labeled: "Local Diet" and accounted for 15.3%, while factor 3 was characterized with, root tuber and animal products, labeled "Energy Boost Diet" and accounted for 15.0%. The factor loading scores for each of these identified factors were divided into quintiles [26,34], (Q1–Q5) for the purpose of establishing their impact on BP status of the respondents.

In spite of the variation noticed in the number of respondents in each of the quintiles, and in the resultant factor scores, the differences were not statistically significant. This signifies that the dietary patterns of civil servants in Bida and Wushishi communities were not significant factors to explain the reasons for variations in BP status. However, the result obtained for BMI ($\chi^2$ = 21.740, $p$-values<0.000) indicated that it was significantly associated with BP status.

### 3.2 Relationships between individual and workplace factors with BP status

Table 2 shows the results of the relationships between individual factors (consisting of socioeconomic, demographics, lifestyle, and food habits), workplace factors and BP status. The highest prevalence rate of HBP was recorded among the Local Government Employees

**Table 1. Distribution and relationship between anthropometric, dietary patterns and BP status.**

| Variables | N (%) | BP Status | | Chi-Square (*p*-values) |
|---|---|---|---|---|
| | | NBP | HBP | |
| **BP Status: Overall** | 476 (100) | 268 (56.3%) | 208 (43.7%) | |
| **Body Mass Index** | | | | 21.740 (0.000) |
| **Under weight** | 15 (3.17) | 10 | 5 | |
| **Normal Weight** | 188 (39.7) | 129 | 59 | |
| **Over Weight** | 156 (32.8) | 79 | 76 | |
| **Obese** | 115 (24.3) | 50 | 65 | |
| **Efficient Diet Pattern** | 216 | | | 0.751 (0.945) |
| Q1 | 43 (19.9) | 21 | 22 | |
| Q2 | 43 (19.9) | 24 | 19 | |
| Q3 | 44 (20.4) | 25 | 19 | |
| Q4 | 43 (19.9) | 22 | 21 | |
| Q5 | 43 (19.9) | 23 | 20 | |
| **Local Diet Pattern** | **225** | | | 4.169 (0.384) |
| Q1 | 46 (20.4) | 23 | 23 | |
| Q2 | 44 (19.6) | 27 | 17 | |
| Q3 | 49 (21.8) | 23 | 26 | |
| Q4 | 41 (18.2) | 26 | 15 | |
| Q5 | 45 (20.0) | 22 | 23 | |
| **Energy Boost Diet Pattern** | **309** | | | 6.443 (0.168) |
| Q1 | 62 (20.1) | 33 | 29 | |
| Q2 | 63 (20.4) | 35 | 28 | |
| Q3 | 62 (20.1) | 26 | 36 | |
| Q4 | 61 (19,7) | 37 | 24 | |
| Q5 | 61 (19.7) | 38 | 23 | |

NBP = Normal BP; HBP = High BP; * = *p*<0.05; Q = Quintile

(51.6%), followed by state employees (43%) which were slightly below the overall average. There were more state civil servants (42%) out of the 476 that responded to the survey, followed by federal (31%) and 27% from Local Government Area. We had more male respondents (311) than females (164). Those who practiced Islam were more numerous than those of other faiths. The ratio of participants in rural to urban was 2 to 3 with more people (35%) receiving monthly income between 18000 & 50000 (in Nigerian currency). The analysis also captured more participants from the Education sector. The mean age of those who participated in the survey was 40.97±9.55 years such that the blood pressure increased significantly ($\chi^2$ = 34.9, p<0.000) with the respondent's age. The MDAs respondents worked in ($\chi^2$ = 18.6, *p*<0.002) were significantly associated with HBP. Tier of workforce, gender, place of residence, income and education status were not statistically significantly associated with HBP among the participants.

Interestingly, the distribution of participants along lifestyle and food habits showed that smoking status, forms of exercise, frequency, and duration of the exercise were not significantly associated with BP status. Only 2% of respondents were current smokers, while about 85% reported to had never smoked, and 12% disclosed were ex-smokers. Most of the respondents (56%) took to trekking as their form of exercise, followed by jogging (29%). Sixty percent of the civil servants participated in exercise to keep fit. On food habits, the findings also showed that 40% of the respondents did not take dinner, while the same proportion stated that

**Table 2. Relationships between individual and workplace characteristics with BP status.**

| Variables | N (%) | BP Status | | Chi-Square (*p*-values) |
|---|---|---|---|---|
| | | NBP | HBP | |
| **Tier of Workplace** | | | | **5.3233** (0.070) |
| Federal | 148 (31.09) | 92 | 56 | |
| State | 200 (42.02) | 114 | 86 | |
| Local Government | 128 (26.89) | 62 | 66 | |
| **Gender** | | | | **3.3964** (0.065) |
| Male | 311 (65.47) | 166 | 145 | |
| Female | 164 (34.53) | 102 | 62 | |
| **Age of Respondents** | | | | **34.900** (0.000)* |
| 15–24 years | 16 (3.40) | 14 | 2 | |
| 25–34 years | 109 (23.14) | 75 | 34 | |
| 35–44 years | 159 (33.76) | 95 | 64 | |
| 45–54 years | 151 (32.06) | 73 | 78 | |
| 55 years | 36 (7.64) | 8 | 28 | |
| **Place of Residence** | | | | **1.1757** (0.278) |
| Rural | 188 (39.75) | 100 | 88 | |
| Urban | 285 (60.25) | 166 | 119 | |
| **Monthly Income Status** | | | | **3.3489** (0.341) |
| Below 18000 | 71 (16.55) | 40 | 31 | |
| 18000–50000 | 152 (35.43) | 91 | 61 | |
| 50001–100000 | 136 (31.70) | 70 | 66 | |
| Above 100000 | 70 (16.32) | 34 | 36 | |
| **MDA Place of Work** | | | | **18.633** (0.002)* |
| Education | 291 (62.85) | 146 | 145 | |
| Health | 75 (16.20 | 54 | 21 | |
| Finance | 16 (3.46) | 7 | 9 | |
| Agriculture | 37 (7.99) | 28 | 9 | |
| Rural Development | 18 (3.89) | 10 | 8 | |
| Others | 26 (5.62) | 15 | 11 | |
| **Education Status** | | | | **0.4708** (0.790 |
| No Education | 8 (1.79) | 4 | 4 | |
| Primary Education | 16 (3.59) | 10 | 6 | |
| Secondary Education+ | 422 (94.62) | 230 | 192 | |
| **Smoking Status** | | | | **0.1464** (0.929) |
| Current Smoker | 8 (1.78) | 4 | 4 | |
| Ex-smoker | 56 (12.44) | 32 | 24 | |
| Never smoke | 386 (85.78) | 218 | 168 | |
| **Frequency of Smoking** | | | | **0.1403** (0.815) |
| 1 or 2 per week | 6 (1.40) | 4 | 2 | |
| 3 or 4 times per week | 34 (7.93) | 21 | 13 | |
| Not at all | 389 (90.68) | 224 | 165 | |
| **Forms of Exercise** | | | | **3.5189** (0.318) |
| Trekking | 231 (55.66) | 126 | 105 | |
| Jogging | 122 (29.40) | 71 | 51 | |
| Swimming | 22 (5.30) | 16 | 6 | |
| Others | 40 (9.64) | 20 | 20 | |
| **Duration of Exercise** | | | | **3.9453** (0.413) |

(*Continued*)

**Table 2.** (Continued)

| Variables | N (%) | BP Status | | Chi-Square (*p*-values) |
|---|---|---|---|---|
| | | NBP | HBP | |
| Less than 10 minutes | | 22 | 15 | |
| 10–20 minutes | | 51 | 45 | |
| 21–30 minutes | | 72 | 67 | |
| 31mins– 1 hour | | 48 | 41 | |
| Above 1 hour | | 45 | 24 | |
| **Meal usually skipped** | | | | **2.4591** (0.483) |
| Breakfast | 104 (24.76) | 54 | 50 | |
| Lunch | 85 (20.24) | 52 | 33 | |
| Dinner | 167 (39.76) | 94 | 73 | |
| None | 64 (15.24) | 32 | 32 | |
| **Reasons for skipping meal** | | | | **9.2787** (0.026)* |
| Limited food | 39 (14.66) | 19 | 20 | |
| No time to prepare it | 78 (29.32) | 46 | 32 | |
| Just a lifestyle | 106 (39.85) | 64 | 42 | |
| To lose weight | 43 (16.17) | 15 | 28 | |
| **Frequency of Eating from Fast Food Joints** | | | | **16.403** (0.006)* |
| Daily | 107 (23.62) | 64 | 43 | |
| 4–6 times a week | 36 (7.95 | 29 | 7 | |
| 2–3 times a week | 11 (2.43) | 6 | 5 | |
| Once a week | 236 (52.10) | 127 | 109 | |
| Occasionally | 35 (7.73) | 20 | 15 | |
| Never | 28 (6.18) | 9 | 19 | |

* = p-value< 0.05,

NHBP = Normal Blood Pressure, HBP = High Blood Pressure

"insufficient time to prepare meal" was the reason for skipping meals. Twenty-four percent ate daily in the food joints. The analysis further revealed that the reasons for skipping meals (Chi-Square = 9.2787, p-values<0.026), frequency of eating meal in restaurants and fast food joints (Chi-Square = 16.40, p-values<0.006) were significantly associated with BP status.

### 3.3 Logistic regression

Table 3 displays the results of logistic regression which was performed in four models to assess the predictors of HBP among civil servants in Bida and Wushishi communities. All the factors that were significantly associated with HBP (including the principal predictor, dietary pattern) were assessed independently in *model 1*. The overall *p*-values for the model were significant, except for the three dietary patterns.

However, the predictors of HBP included: obesity (OR = 2.600, 95% CI = 0.84 to 8.090); being more than 34 years of age: 35–44 years (OR = 4.716, 95% CI = 1.04 to 21.5), 45–54 years (OR = 7.48, 95% CI = 1.64 to 34.05) and 55 years + (OR = 24.5, 95% CI = 4.6 to 131.1); eating often in fast food joints or restaurants (4–6 times per week), and work in the education sector. These odd ratios (ORs) obtained implied that obese subjects had 2.6 times more likelihood of having HBP than someone who was underweight (reference category). In the same way, a respondent who was 55 years and above was 24 times more likely to have hypertension than one who was 15–24 years (reference category). A participant who worked in either Health or

**Table 3. Logistic regression analysis (unadjusted & adjusted) of BP status.**

| Variables | UOR (95% CI) (Model1) | AOR (Model2) | AOR (Model3) | AOR (Model4) |
|---|---|---|---|---|
| **Efficient Diet Pattern** | O. p-values 0.944 | **0.0060***  | **0.0002***  | **0.0038***  |
| Q1 | 1.000(ref) | 1.000 | | |
| Q2 | 0.756(0.32 1.765) | 0.324 | | |
| Q3 | 0.725(0.31 1.688) | 0.460 | | |
| Q4 | 0.911(0.39 2.122) | 0.941 | | |
| Q5 | 0.830(0.35 1.935) | 0.083* | | |
| **Local Diet Pattern** | O. p-values 0.378 | | 1.000 | 1.000 |
| Q1 | 1.000(ref) | | 0.757 | 1.143 |
| Q2 | 0.630(0.27 1.456) | | 1.210 | 1.967 |
| Q3 | 1.130(0.51 2.530) | | 0.551 | 0.726 |
| Q4 | 0.577(0.24 1.362) | | 2.058 | 0.579 |
| Q5 | 1.045(0.46 2.378) | | 1.000 | 1.000 |
| **Energy Boost Diet Pattern** | O. p-values 0.168 | | | |
| Q1 | 1.000(ref) | | | |
| Q2 | 0.910(0.45 1.841) | | | |
| Q3 | 1.576(0.78 3.203) | | | |
| Q4 | 0.738(0.36 1.510) | | | |
| Q5 | 0.689(0.34 1.41) | | | |
| **Body Mass Index** | O. p-values 0.000* | | | |
| Under weight | 1.00(Ref) | 1.000 | 1.000 | 1.000 |
| Normal Weight | 0.915(0.30 2.80) | 0.215 | 0.065 | 0.262 |
| Over Weight | 1.924(0.63 5.89) | 0.637 | 0.524 | 0.950 |
| Obese | 2.600(0.84 8.09)* | 3.153 | 0.480 | 1.710 |
| **MDA of Work Place** | O. p-values 0.002* | | | |
| Education | 1.000(Ref) | 1.000 | 1.000 | 1.000 |
| Health | 0.392(0.22 0.68)* | 0.480 | 0.300 | 0.872 |
| Finance | 1.295(0.47 3.56) | 1.160 | 0.801 | 0.478 |
| Agriculture | 0.324(0.14 0.71)* | 0.263 | 0.333 | 0.296 |
| Rural development | 0.806(0.31 2.10) | 0.279 | 0.130 | 1.340 |
| Others | 0.738(0.32 1.66) | - | 1.789 | 3.773 |
| **Age in Groups** | O. p-values 0.000* | | | |
| 15–24 years | 1.000(ref) | 1.000 | 1.000 | 1.000 |
| 25–34 years | 3.173(0.68 14.74) | 1.822 | 1.765 | 0.832 |
| 35–44 years | 4.716(1.04 21.5)* | 6.077 | 5.320 | 2.383 |
| 45–55 years | 7.48(1.64 34.05)* | 6.343 | 4.416 | 1.630 |
| 55 years+ | 24.5(4.6 131.07)* | 1.000 | 19.93 | - |
| **Frequency of eating in Restaurants/Fast Food Joints** | O. p-values 0.004* | | | |
| Daily | | 1.000 | 1.000 | 1.000 |
| 4–6 times a week | | 0.610 | 0.488 | 0.327 |
| 2–3 times a week | 1.000(ref) | - | 0.188 | 0.260 |
| Once a week | 0.36 (0.144 0.89)* | 1.440 | 1.427 | 2.220 |
| Occasionally | 1.24(0.356 4.321) | 2.977 | 10.12 | 2.668 |
| Never | 1.277(0.80 2.031) | 0.309 | 0.732 | 2.770 |
| **Reasons for Skipping Meals** | O. p-values 0.025* | | | |
| Limited food | 1.000 (ref) | 1.000 | 1.000 | 1.000 |
| No time to prepare | 0.661(0.31 1.432) | 0.493 | 0.434 | 1.690 |

*(Continued)*

**Table 3.** (Continued)

| Variables | UOR (95% CI) (Model1) | AOR (Model2) | AOR (Model3) | AOR (Model4) |
|---|---|---|---|---|
| My lifestyle | 0.623(0.30 1.305) | 0.490 | 0.633 | 0.870 |
| To lose weight | 1.773(0.73 4.307) | 1.005 | 2.886 | 1.747 |

O.p-values is the overall *p*-values for the significant of the model,

\* = $p < 0.05$,

UOR = unadjusted odd ratios, AOR = adjusted odd ratios and ref = reference category

Agricultural sectors had 0.6 and 0.7, respectively, less likelihood to have hypertension than one who worked in education sector (reference category). *Models 2–4* considered logistic regression for the three dietary patterns and adjusted for confounding factors (i.e. all factors earlier considered to be significant factors), none were found as predictor of hypertension although the overall *p*-values indicated that all the models were significant. However, in model 2 after adjusting for all confounding factors, being in the quintile 5 (OR = 0.083, p<0.009) became a statistically significant predictor of HBP. This result showed that, having a high score in "Efficient Diet Pattern" had 0.92 less likelihood to be hypertensive than the one in quintile 1 (lowest score-reference category).

## 4 Discussion

In this study, a high prevalence of hypertension (44%) was found among the civil workforce in Bida and Wushishi communities of Niger state, Nigeria. This relatively high prevalence of HBP agrees with results obtained from similar studies[41,42] but differs with other findings [5,11]. These variations may be attributed to the differences in studied populations. Increased age was also identified as a risk factor in this study where participants above the age of 34 years were more likely to be hypertensive than those below the age of 25 years, in agreement with several other studies[11,12], but contrary to other findings in Gurven et al[43]. The Ministry, Department and Agency (MDA) where the participants worked was statistically significantly associated with HBP. Those who worked in Health and Agricultural sectors were found to be less likely to be hypertensive than those who worked in education sector. The possible reason may be that those in the Health sector were more likely to have access to prompt health care services and those in Agriculture sector are more likely to have access to good food. Non-regular payment of workers' salaries is more common among staff of education sector in Nigeria and this may affect their purchasing power to eat good foods. Like many other studies [5,11], BMI was found to be a risk factor for HBP. The possible reason is that most civil servants live a sedentary lifestyle [20]. Participants who ate only 4–6 times a week in restaurants/fast food joints were 0.64 less likely to be hypertensive than those who ate outside daily. This study also found that smoking status among civil workforce was not a significant risk factor for HBP, contrary to the findings from another study [11], but in agreement with some other observational studies [43–45]. Three different dietary patterns were identified in this study: "Efficient Diet," "Local Diet," and "Energy Boost Diet." None of these were statistically significant factors for HBP. However, after adjusting for other variables, the highest quintile score (Q5) in "efficient diet" became significant with 0.92 less likelihood to be hypertensive than someone with the lowest quintile score (Q1). The reason being that a higher score in combination with food groups characterized by vegetables, fruits, and plant protein is expected to have a lower risk of HBP[23,24,30].

## 5 Implications and limitations to study

The implications for this study are that civil servants who are 35 years old and above should always have their BP status checked on a regular basis. A sedentary lifestyle should be avoided with a call on civil servants to participate actively in regular exercising. However, the interpretation of the results from this study is limited to the fact that this study has a cross-sectional design and biochemical samples were neither collected nor analyzed and hence causality could not established. Also, there were no multiple measures of BP taken except when SBP was outside the range of 80-mm Hg and 240-mm Hg and DBP were less than 30-mm Hg, in which case the measure was repeated after a 5 minutes interval to validate the reading. In view of the inconsistencies in responses (decline to disclose), to a section of the instrument relating to history of health status, no participant was excluded on the basis of having health status that could modify blood pressure. Access to the actual number of civil servants in the three tiers of workplace in this area of study was difficult, so survey design was not gender-stratified. However, a ratio of 2 to 1 for male to female was used in line with a previous study[20] performed in this study area. From the above, the authors submit that restrain could be exercised in generalizing the results to other parts of Niger State and Nigerian civil workforce. However, the findings among this subset of civil workforce are quite important to undertake large scale assessment of the impact of variables considered in this study on blood pressure across the civil workforce in Niger state and Nigeria.

## 6 Conclusions

This research work is a survey type designed to investigate the impact of dietary pattern, individual and workplace characteristics on hypertension among civil servants in Bida and Wushishi communities of Niger State, Nigeria. The findings from this study have identified individual anthropometric, socioeconomic, demographic, and workplace characteristics that could predict the likelihood of having hypertension among civil servants in Niger State. These, in turn, can enhance informed decision making for adequate formulation and implementation of policies to improve the general health and wellbeing of civil servants, and also at eliminating inhibiting factors in similar communities toward attaining WHO target for the country.

## Supporting information

**S1 Data. Used dataset for ibr 2018.**
(XLSX)

## Acknowledgments

The authors acknowledged the support given to them by the Rector and the management of Niger State Polytechnic; the Administrative Heads of the institutions visited for data collection, Ogala and Bake for initial proof reading of the draft manuscript. We also acknowledged Editage team for language editing, and the supporting staff, especially the enumerators/coding clerks: Wonder-Lois, Tega and Abdul.

## Author Contributions

**Conceptualization:** Phillips Edomwonyi Obasohan, Josephine N. Okorie.

**Data curation:** Phillips Edomwonyi Obasohan, Josephine N. Okorie.

**Formal analysis:** Phillips Edomwonyi Obasohan.

**Funding acquisition:** Phillips Edomwonyi Obasohan, Josephine N. Okorie, Adamu Lapai Sule, Khadijat Jumai Ndako.

**Investigation:** Phillips Edomwonyi Obasohan, Josephine N. Okorie, Adamu Lapai Sule, Khadijat Jumai Ndako.

**Methodology:** Phillips Edomwonyi Obasohan.

**Project administration:** Phillips Edomwonyi Obasohan, Josephine N. Okorie, Adamu Lapai Sule, Khadijat Jumai Ndako.

**Software:** Phillips Edomwonyi Obasohan.

**Supervision:** Phillips Edomwonyi Obasohan, Josephine N. Okorie, Adamu Lapai Sule, Khadijat Jumai Ndako.

**Validation:** Phillips Edomwonyi Obasohan, Josephine N. Okorie.

**Writing – original draft:** Phillips Edomwonyi Obasohan.

**Writing – review & editing:** Phillips Edomwonyi Obasohan, Josephine N. Okorie, Adamu Lapai Sule, Khadijat Jumai Ndako.

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
