## [Decision Letter · Decision Letter 0]

5 Sep 2019

PONE-D-19-20073

The Impact of Dietary Patterns, Individual and Workplace Characteristics on Blood Pressure status among Civil Servants in Bida and Wushishi Communities of Niger State, Nigeria: Factor and Logistic Regression Analyses Approach

PLOS ONE

Dear Mr Obasohan,

Thank you for submitting your manuscript to PLOS ONE. After careful consideration, we feel that it has merit but does not fully meet PLOS ONE’s publication criteria as it currently stands. Therefore, we invite you to submit a revised version of the manuscript that addresses the points raised during the review process.

ADDITIONAL ACADEMIC EDITOR COMMENTS: An expert in this field handled your manuscript and found it interesting, but they have listed several comments and concerns that need to be addressed. These comments include, but are not limited to, the need to provide a better rationale in the introduction about the study design and there seems to be some methodological details lacking in the methods section.

We would appreciate receiving your revised manuscript by Oct 20 2019 11:59PM. To enhance the reproducibility of your results, we recommend that if applicable you deposit your laboratory protocols in protocols.io, where a protocol can be assigned its own identifier (DOI) such that it can be cited independently in the future. For instructions see: http://journals.plos.org/plosone/s/submission-guidelines#loc-laboratory-protocols

We look forward to receiving your revised manuscript.

Kind regards,

Frank T. Spradley

Academic Editor

PLOS ONE

Journal Requirements:

2. Please provide more information regarding the setting (e.g. locations, relevant dates, periods of recruitment, data collection) as well as the sources, methods and criteria of participants' selection

4. We note you have included a table to which you do not refer in the text of your manuscript. Please ensure that you refer to Tables 2 and 3 in your text; if accepted, production will need this reference to link the reader to the Table.

Reviewers' comments:

Reviewer's Responses to Questions

**Comments to the Author**

1. Is the manuscript technically sound, and do the data support the conclusions?

Reviewer #1: Yes

2. Has the statistical analysis been performed appropriately and rigorously? 

Reviewer #1: Yes

3. Have the authors made all data underlying the findings in their manuscript fully available?

Reviewer #1: Yes

4. Is the manuscript presented in an intelligible fashion and written in standard English?

Reviewer #1: No

5. Review Comments to the Author

Reviewer #1: Title section: probably would need to exclude "Factor and logistic regression analyses"

This study used a cross sectional study, but as the title indicates, the study tried to assess the impact of some variables. Is that right? Please modify your title.

Abstract section: It is recommended not to use abbreviations in this section. In addition, you said that this problem has not been explored in Nigeria, how can you be sure that?

Take home message should be added in conclusion

Introduction section:

1. Page 3 (Line 65, 72), please correct as SSA, provide their full text first for SBP and DBP

2. Page 3, 4. Please rewrite the sentence beginning with between blood pressure and end with BMI (Line, 82, 83)

3. Page 4, Line 91. Where is elsewhere?

4. Page 5, Line 102-107. Taking about factor analyses, I think this paragraph should be stated in methodology section

5. In general, when we state the introduction section, we have to state in the following way;

6. General description of the problem under study including its meaning, burden/magnitude of the problem from global to local and from past to current, whose is affected, where, when, the consequences of the problem, causes, what has been done before by the Government of Nigeria to reduce this problem, factors attributable of the problem identified by previous research works.

Methods section:

1. Page 5, Line 115. Description of the study setting and design were not included

2. Page 6, Line 122. Describing about DDF is better if you move it in to variables measurement and operational definition section

3. Page 7. You estimated a final sample size of 440, then, why you included 530 participants. Is your sampling technique being cluster sampling, if so, you need to account the design effect while calculating your sample size?

4. Many sections of the methodology were not appeared. Sampling procedure, study population, exclusion and inclusion criteria, variable measurements, operational definition, and others. Please follow the scientific paper writing format as others did.

5. Where were your data collectors, their profession and educational background, the language used to prepare the data collection tool….

Results section:

1. Page 10. You did not state the results of sociodemographic and other variables. No interest,..

2. Define others in respective tables (Table, 1, 2).

6. PLOS authors have the option to publish the peer review history of their article (what does this mean?). If published, this will include your full peer review and any attached files.

Reviewer #1: No

---

## [Decision Letter · Decision Letter 1]

22 Nov 2019

Impact of Dietary Patterns, Individual and Workplace Characteristics on Blood Pressure status among Civil Servants in Bida and Wushishi Communities of Niger State, Nigeria

PONE-D-19-20073R1

Dear Dr. Obasohan,

We are pleased to inform you that your manuscript has been judged scientifically suitable for publication and will be formally accepted for publication once it complies with all outstanding technical requirements.

With kind regards,

Frank T. Spradley

Academic Editor

PLOS ONE

Reviewers' comments:

Reviewer's Responses to Questions

**Comments to the Author**

1. If the authors have adequately addressed your comments raised in a previous round of review and you feel that this manuscript is now acceptable for publication, you may indicate that here to bypass the “Comments to the Author” section, enter your conflict of interest statement in the “Confidential to Editor” section, and submit your "Accept" recommendation.

Reviewer #1: All comments have been addressed

2. Is the manuscript technically sound, and do the data support the conclusions?

Reviewer #1: Yes

3. Has the statistical analysis been performed appropriately and rigorously? 

Reviewer #1: Yes

4. Have the authors made all data underlying the findings in their manuscript fully available?

Reviewer #1: Yes

5. Is the manuscript presented in an intelligible fashion and written in standard English?

Reviewer #1: Yes

6. Review Comments to the Author

Reviewer #1: The manuscript has been improved a lot, I have no additional comments for the author, including concerns about dual publication, research ethics, or publication ethics to be added.

7. PLOS authors have the option to publish the peer review history of their article (what does this mean?). If published, this will include your full peer review and any attached files.

Reviewer #1: No

---

## [Editor Report · Acceptance letter]

6 Dec 2019

PONE-D-19-20073R1 

Impact of Dietary Patterns, Individual and Workplace Characteristics on Blood Pressure status among Civil Servants in Bida and Wushishi Communities of Niger State, Nigeria 

Dear Dr. Obasohan:

I am pleased to inform you that your manuscript has been deemed suitable for publication in PLOS ONE. Congratulations! Your manuscript is now with our production department. 

With kind regards,

on behalf of

Dr. Frank T. Spradley 

Academic Editor

PLOS ONE